# Topological structures and syntenic conservation in sea anemone genomes

Bob Zimmermann [1,2,7], Juan D. Montenegro[1,2,7], Sofia M. C. Robb [3], Whitney J. Fropf[3], Lukas Weilguny[1], Shuonan He[3], Shiyuan Chen [3], Jessica Lovegrove-Walsh [1], Eric M. Hill[3], Cheng-Yi Chen[3], Katerina Ragkousi[3,4], Daniela Praher[1], David Fredman[1], Darrin Schultz [1], Yehu Moran [1,6], Oleg Simakov [1,2], Grigory Genikhovich [1], Matthew C. Gibson [3] ✉ & Ulrich Technau [1,2,5] ✉

There is currently little information about the evolution of gene clusters, genome architectures and karyotypes in early branching animals. Slowly evolving anthozoan cnidarians can be particularly informative about the evolution of these genome features. Here we report chromosome-level genome assemblies of two related anthozoans, the sea anemones *Nematostella vectensis* and *Scolanthus callimorphus*. We find a robust set of 15 chromosomes with a clear one-to-one correspondence between the two species. Both genomes show chromosomal conservation, allowing us to reconstruct ancestral cnidarian and metazoan chromosomal blocks, consisting of at least 19 and 16 ancestral linkage groups, respectively. We show that, in contrast to Bilateria, the Hox and NK clusters of investigated cnidarians are largely disintegrated, despite the presence of staggered *hox/gbx* expression in *Nematostella*. This loss of microsynteny conservation may be facilitated by shorter distances between cis-regulatory sequences and their cognate transcriptional start sites. We find no clear evidence for topologically associated domains, suggesting fundamental differences in long-range gene regulation compared to vertebrates. These data suggest that large sets of ancestral metazoan genes have been retained in ancestral linkage groups of some extant lineages; yet, higher order gene regulation with associated 3D architecture may have evolved only after the cnidarian-bilaterian split.

Comparative genomics and epigenomics have provided fundamental insights into the evolution of gene regulation in multicellular organisms. Among basally branching animals, this led to the surprising discovery of conserved gene repertoires[1–3], of microsyntenic gene clusters and patterns of DNA methylation and histone modification codes[4–6]. Recent applications of long-read sequencing technologies and proximity ligation methods such as Hi-C facilitated the genome assembly on chromosomal level of many organisms. These chromosome-level genome assemblies have presented the opportunity to compare the content and localization of homologous genes

[1]Department of Neurosciences and Developmental Biology, Faculty of Life Sciences, University of Vienna, Djerassiplatz 1, 1030 Vienna, Austria. [2]Research platform SinCeReSt, University of Vienna, Djerassiplatz 1, 1030 Vienna, Austria. [3]Stowers Institute for Medical Research, Kansas City, MO 64110, USA. [4]Department of Biology, Amherst College, Amherst, MA 01002, USA. [5]Max Perutz laboratories, University of Vienna, Dr. Bohrgasse 5, 1030 Vienna, Austria. [6]Present address: The Alexander Silberman Institute of Life Sciences, Faculty of Science, The Hebrew University of Jerusalem, Jerusalem 9190401, Israel. [7]These authors contributed equally: Bob Zimmermann, Juan D. Montenegro. ✉e-mail: mg2@stowers.org; ulrich.technau@univie.ac.at

between distantly related species and the reconstruction of ancestral linkage groups. In vertebrates and several other bilaterians, topological analysis has revealed three-dimensional chromosomal architecture[7], organized as chromosomal compartments and topologically associated domains (TADs)[8]. The boundaries of TADs have been shown to act as barriers of gene regulation[9–12]. Analyses on chromosome evolution and three-dimensional genome structure have mainly focused on bilaterians, and it remains unclear whether non-bilaterian animals share the topological signatures of bilaterian chromosomes. In this regard, studying Cnidaria, the sister clade to Bilateria, is a crucial step to understanding the evolution of animal genomes.

Cnidaria constitutes a large clade of basally branching Metazoa, dating back between 590 and 690 Mya[13–15]. Their robust phylogenetic position as the sister group to Bilateria makes them the key group to study the evolution of bilaterian features, such as axis organization,

mesoderm formation and central nervous system development[16]. The starlet sea anemone, *Nematostella vectensis* (Fig. 1a−Hexacorallia; Actiniaria; Edwardsiidae), has been developed into an important model organism[17–19] and in 2007 became the first non-bilaterian animal to have a draft scaffold-level genome assembled[1]. The *Nematostella* genome revealed uncanny conservation of gene content to vertebrates as well as the first observations of large-scale macrosyntenic conservation by way of comparisons with human chromosomes to *Nematostella* scaffolds[1].

Since the release of the *Nematostella* genome, genomes of the representatives of all five cnidarian classes have become available[2,20–27] providing valuable insight into various aspects of the cnidarian gene complement and genome organization. However, these genomes originated from distantly-related species, and few genomic studies of cnidarians have sought to search for genomic

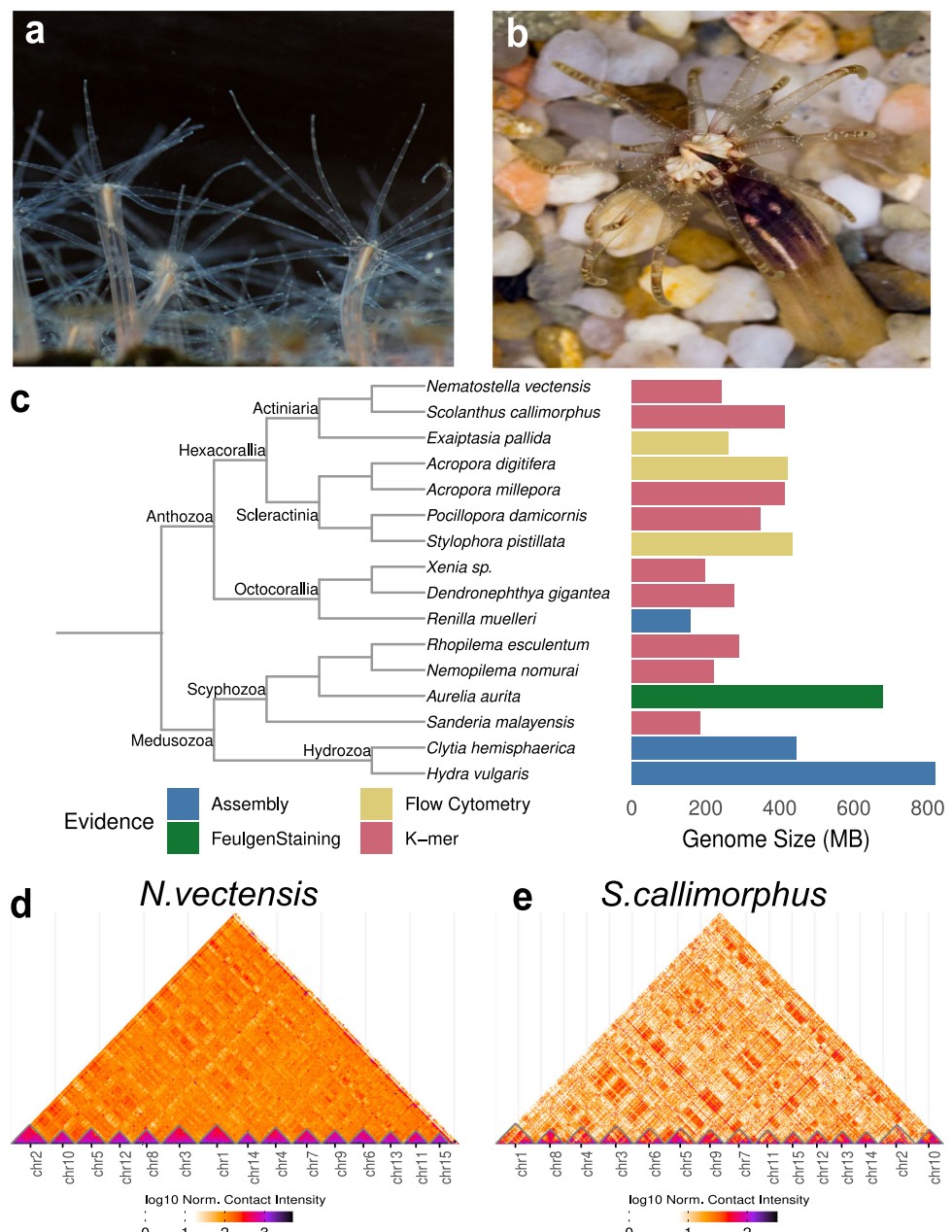

**Fig. 1 | Chromosomal assembly of edwardsiid sea anemones *Nematostella vectensis* and *Scolanthus callimorphus*. a** *Nematostella vectensis*. photo credit: Patrick RH Steinmetz. **b** *Scolanthus callimorphus*. photo credit: Robert Reischl. **c** Genome size estimates of cnidarian genomes taken from refs. 2,21,22,24−26,32,39,132,155−158. Contact maps of *N. vectensis* scaffolds (**d**) and *S. callimorphus* contigs (**e**).

conservation signals of recently-diverged species, a cornerstone of comparative genomics[28]. For example, no genome sequence of an edwardsiid sea anemone has been reported, aside from that of *Nematostella*. Another sea anemone in the family Edwardsiidae is the "worm sea anemone" *Scolanthus callimorphus* (Fig. 1b), dwelling in European intertidal zones[29,30], which according to our molecular clock calculations has separated from *Nematostella vectensis* ~174 Mio years ago, to date its closest sequenced relative (Supplementary Fig. 1, see Materials and Methods for details).

Here, we report chromosome-level genome assemblies and new gene models for the model organism *Nematostella vectensis* and of the closely-related species *Scolanthus callimorphus*. We show that the macrosyntenic localization of orthologous genes across the chromosomes are highly conserved across many cnidarians and bilaterians, allowing us to reconstruct the minimal set of ancestral eumetazoan chromosomal linkage blocks. Notably, the macrosyntenic conservation contrasts with the loss of most of the microsyntenic blocks, including the integrity of the Hox cluster. Moreover, in *Nematostella*, which like all non-bilaterians lacks the TAD boundary protein CTCF[4,31], HiC analyses did not reveal any obvious TAD-like structures, as known in Bilateria. We propose that the proximity of most cis-regulatory regions to their target genes is an ancestral metazoan feature, whereas large-scale three-dimensional structures, such as TADs, only became necessary in bilaterians with larger genomes and more distantly located enhancers.

## Results

### High quality chromosome-level assemblies of two Edwardsiid genomes

Using short-read sequencing and a *k*-mer coverage model, we estimated the genome length of *Nematostella* at 244 Mb (Supplementary Fig. 2), which is substantially shorter than previously suggested at 450 Mb[1]. This discrepancy could be partly attributed to the previous use of four haplotypes in sequencing, lower coverage and read length. The genome of the sea anemone *Exaiptasia pallida* is similar in length to *Nematostella*[32], while the estimated 414 Mb of the *Scolanthus* genome is at present the largest sequenced sea anemone genome (Fig. 1c). Using PacBio long-read sequencing and high-throughput conformation capture (Hi-C), we then assembled chromosome-level *Nematostella* and *Scolanthus* genomes, which surpass the quality of the original *Nematostella* genome in terms of contiguity, accuracy and mappability (see Supplementary Notes, Supplementary Figs. 2,3 for details).

Gene completeness as measured by alignment of single-copy pan-metazoan genes to the assemblies using the BUSCO method[33] was comparable to the previous assembly (Supplementary Fig. 2i), however in order to completely assess this, we generated new gene models. Using a combination of IsoSeq and RNAseq data, we identified 24,525 gene models and 36,280 transcripts (see Supplementary Notes for details). BUSCO analysis showed that the gene set was more complete than a previously-generated gene set[34]. In addition, the new gene models showed a better mappability to a publicly available single-cell RNA-Seq data set[35,36] (Supplementary Fig. 4).

To facilitate the usage of the newly assembled genomes, we established a publicly accessible genome browser. Both new genome assemblies and associated data are available for browsing, downloading, and BLAST at SIMRbase (https://simrbase.stowers.org). The *Nematostella vectensis* genome assembly, referred to as Nvec200, has an abundance of aligned track data, including the newly generated gene models, a large collection of published RNAseq and ChIP-seq analyses, as well as 145 ultra-conserved non-coding elements (UCNEs) shared between *Nematostella* and *Scolanthus* (Supplementary Notes; Supplementary data file 8).

### Comparison of *Nematostella* chromosomes to metazoan genomes

We identified 15 chromosomes in the new genome assemblies of *Nematostella* and *Scolanthus*, and we numbered these chromosomes from 1 to 15 according to decreasing size (Fig. 1d, e). This is in line with the previous estimates based on the number of *Nematostella* metaphase plates[1] and the analysis of *Nematostella* chromosome spreads[37]. Most chromosomes, according to their homologous pair, corresponded in length but are much larger in *Scolanthus* (Fig. 4d). This was accounted for by a large fraction of unclassified and potentially lineage-specific repeat sequences (Supplementary data file 1, Supplementary Notes). No obvious heteromorphic sex chromosomes could be identified based on read mapping depth, or from the level of heterochromatin usually associated with non-recombinant sex chromosomes. Ascertaining the sex-determining regions of the genome will require more data and detailed analyses.

Next we wished to determine the extent to which the *Nematostella* and *Scolanthus* chromosomes exhibit conservation of gene content and order (micro- and macrosynteny). Indeed, each of the 15 chromosomes of both species shared a majority of orthologous genes with a single corresponding chromosome in the other species (Fig. 2a). We found that 8117 of 8692 mutual best BLAST hits between *Nematostella* and *Scolanthus* were retained on their respective chromosomes, implying a one-to-one homology between all 15 chromosomes. However, gene order was largely lost from the most recent common ancestor (MRCA), which we estimate to have diverged approximately 174 Mya (Supplementary Fig. 1).

To assess the macrosyntenic conservation between anthozoans we compared the *Nematostella* chromosomes to those of the sea anemone *Exaiptasia pallida*, the stony coral *Acropora millepora* and the soft coral *Xenia sp.* (Fig. 2a). While both *Exaiptasia* and *Acropora* are only assembled on the scaffold level, we observed that the gene content of the scaffolds suggest a similar karyotype to *Nematostella*. The chromosome-level genome assembly of the octocoral soft coral *Xenia sp.* also appears to have 15 chromosome-scale scaffolds[20]. However, despite the apparent identical number of chromosomes, the chromosome-scale scaffolds of *Xenia* showed surprisingly little homology to those of the hexacoral *Nematostella*. We discovered many translocations and fusions, with only three chromosomes demonstrating a one-to-one relationship (Fig. 2a).

Notably, the hydrozoan *Hydra vulgaris*, which belongs to the clade of Medusozoa, also has 15 chromosomes[38]. While our analyses revealed many ancestral linkage groups in the comparison with *Hydra*, we also detected numerous chromosomal rearrangements, including splits and fusions (Fig. 2a, Supplementary Fig. 5). Specifically, we observe that *Nematostella* chromosomes 2 and 3 are partially or completely split in other cnidarian genomes. By comparison, the scyphozoan jellyfish *Rhopilema* has 21 chromosomes, supported with karyotype images[21,39]. We observed that all *Rhopilema* and *Nematostella* chromosomes show a clear 1-to-1, 1-to-2 or, in a single case, a 1-to-3 macrosyntenic correspondence, which can be traced to the ancestral linkage groups (Supplementary Figs. 5, 7). These comparisons among cnidarians suggest that the history of cnidarian chromosomes may be more complex than previously envisioned. This is in line with recent interpretations of cnidarian chromosome comparisons[40].

We then compared the chromosome-scale scaffolds of *Nematostella* with that of a bilaterian, the cephalochordate *Branchiostoma floridae*. Cephalochordates are early branching chordates lacking the two rounds of whole-genome duplication and allotetraploidization suggested for vertebrates[41,42], as represented by their single Hox cluster[43] (Fig. 2a). Strikingly, *Branchiostoma* and *Nematostella* chromosomes retain extensive macrosynteny since their divergence from the bilaterian-cnidarian MRCA. The large-scale macrosynteny retention detected in *Branchiostoma* becomes less obvious once we focus

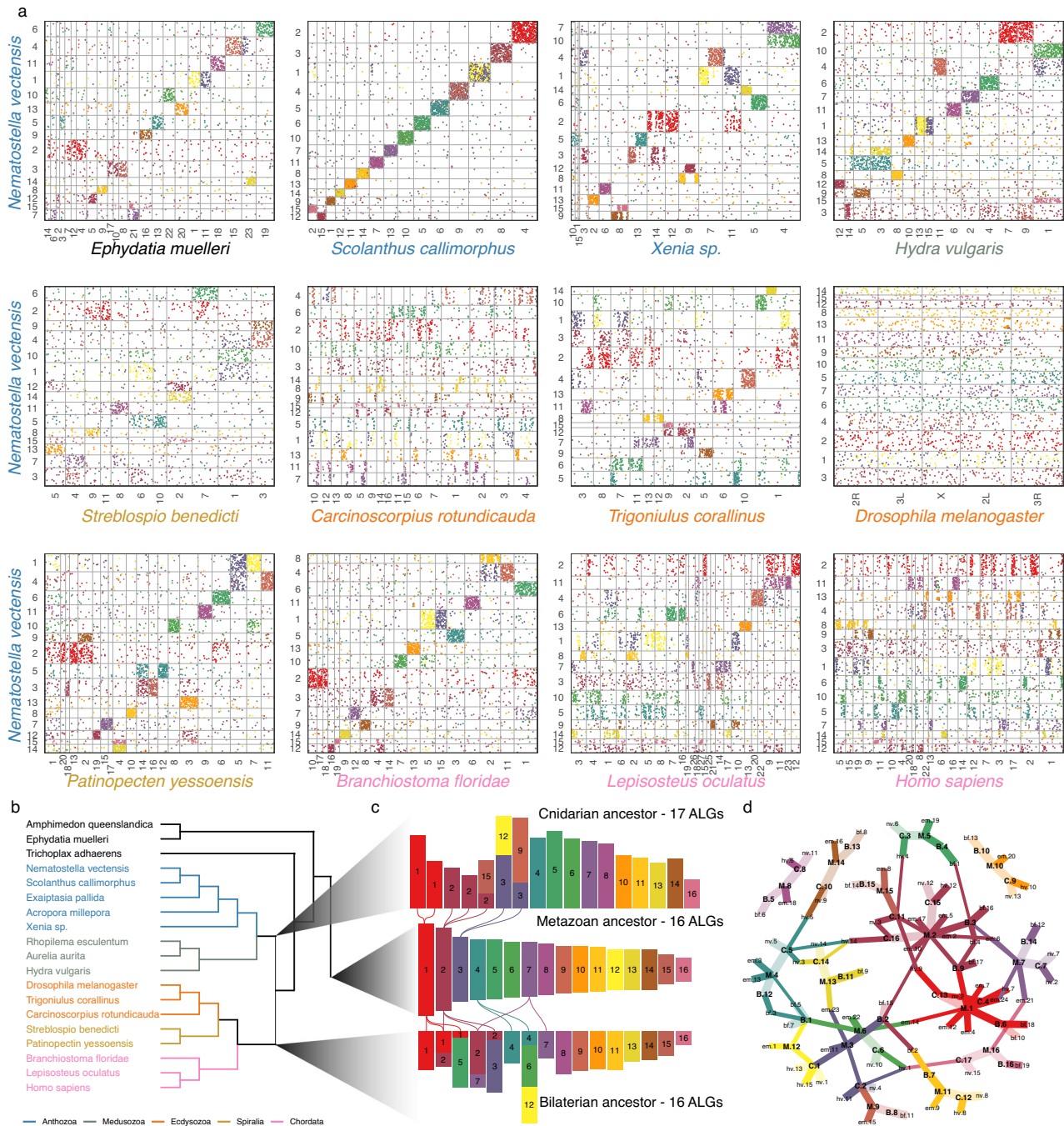

**Fig. 2 | Macrosynteny conservation of edwardsiid genomes and reconstruction of ancestral linkage groups. a** Oxford plots of the macrosyntenic relationships of *Nematostella vectensis* (y-axis) to genomes of various metazoans (x-axis). Pairwise orthologous are ordered by their position on the chromosome labeled on the axis. Dots (genes) are colored according to their membership to the metazoan ancestral linkage groups (ALGs) shown in (**c**). Species names are colored according to clade shown in (**b**). **b** Genomes represented in this figure and Supplementary Figs. 5–7. **c** Relationships between the ALGs colored according to the metazoan ALG. Lines

are drawn between ALGs to represent fissions. **d** Graph representing the relationships of metazoan (M), bilaterian (B), and cnidarian (C) ALGs to each other and extant genomes of *Ephydatia* (em), *Nematostella* (nv), *Rhopilema* (re) and *Branchiostoma* (bf). Edges are colored by the metazoan ALG in (**b**) from which its path has a source. Line width represents the fraction of the extant genome or derived ALG (bilaterian or cnidarian) content of the ancestral genome. Faded edges represent near one-to-one relationships (>0.8).

on more recently-branching bilaterian clades. Vertebrates, such as the early branching actinopterygian fish *Lepisosteus oculatus*[44] and humans, underwent two or more rounds of genome duplications and showed additional translocation events from the MRCA (Fig. 2a). Among protostomes, we observed that the lophotrochozoan Ram's horn worm *Streplosbio benedicti* and the sea scallop *Patinopecten yessoensis*[45], as well as the ecdysozoan millipede *Trigoniulus corallinus*[46], retained many macrosyntenic links (Fig. 2a).

The high degree of macrosynteny conservation with *Branchiostoma* motivated us to compare *Nematostella* chromosomes to those of the representative of an even older lineage, the Porifera (sponges), which branched off prior to the bilaterian-cnidarian split (Supplementary Fig. 1) and is considered by many to be the sister group to all other animals[47,48]. Strikingly, the comparison with the recently-assembled chromosomes of the sponge *Ephydatia muelleri* revealed many shared ancestral linkage groups (Fig. 2a).

## Reconstruction of ancestral linkage groups

The observed patterns of conserved macrosynteny between chromosomes of multiple distantly-related species and the varying degrees of gene shuffling and chromosomal rearrangements in others (e.g., *Drosophila*, *Caenorhabditis*, humans) prompted us to reconstruct the set of predicted ancestral linkage groups for the MRCA of cnidarians, bilaterians and metazoans (Fig. 2b). First, we inferred gene orthology across all genomes investigated. For each of Metazoa, Bilateria and Cnidaria, we determined ancestral linkage groups by determining genes with high chromosomal linkage throughout each clade. Our method finds ALGs which maximize their "modularity", i.e., groups with high chromosomal linkage within ALGs and low linkage between ALGs, by evaluating multiple scenarios (see Methods and Supplemental notes for details) to infer "ancestral linkage groups" (ALGs). We can then color the oxford plots with the genes in these blocks suggesting a pattern of splits and fusions in extant chromosomes from ancestral chromosomes.

We identified 17 ALGs present in the ancestor of cnidarians, although we could find similar modularity in scenarios of 16–18 ALGs (see Supplementary notes for details). These correspond well to the extant cnidarian chromosomes (Fig. 2c, Supplementary Fig. 5), however, while the *Rhopilema* and edwardsiid chromosomes appear highly representative of the ancestral cnidarian karyotype, the soft coral *Xenia* chromosomes appear to have undergone more chromosomal translocation events. Our results also indicate that although both Hydrozoa and Edwardsiidae have a clear 15 chromosome karyotype, their chromosomes originate from distinct fusions of cnidarian ALGs. For the ancestor of Bilateria, we identified 16 ALGs (Fig. 2c). In a previous study 17 ALGs were proposed[49], which resulted in similarly high modularity in our analysis (see Supplementary notes for details). By comparison with the sponge *Ephydatia*, we reconstructed a minimum of 16 metazoan ALGs for the metazoan ancestor, which were maintained in the cnidarian MRCA, Bilateria and the sponge (Fig. 2b, c). We also compare this to a recent reconstruction of the ancestor to bilaterians, cnidarians, and sponges in Supplementary data file 13[40].

In order to visualize how the chromosomes of extant species have undergone major splits and fusions, we projected the reconstructed ALGs of the respective cnidarian, bilaterian and metazoan ancestor to the chromosomes of the extant species. We found that many ALGs correspond to the same chromosomes across multiple species and lineages (Fig. 2c; Supplementary Figs. 5–7). To determine the extent of this, we further explored the relationships between metazoan ALGs and those of cnidarians and bilaterians (Fig. 2c). Remarkably, many ancestral chromosomes exhibited a 1-to-1 correspondence across all predicted ancestral lineages (Fig. 2c) and carried through to the extant lineages (Fig. 2d).

## Chromosomal organization of the NK and extended Hox gene clusters

The chromosome-level assembly of the *Nematostella* genome allowed us to address the evolution of specific gene clusters. Prominent examples of clusters of homeodomain transcription factor coding genes ancestral for Bilateria include the SuperHox cluster, the ParaHox cluster, the NK/NK-like cluster as well as *NK2* group genes located separately[43,50,51]. It has been hypothesized that all of them originated from a single gene cluster, which then disintegrated during evolution[51]. Our analysis revealed that *Nematostella* possesses a separate ParaHox cluster of two genes, (*Gsx* and *Xlox/Cdx*) on chromosome 10, and a SuperHox cluster on chromosome 2 containing *Hox*, *Evx*, *Mnx*, and *Rough*, as well as more distant *Mox* and *Gbx*[52] (Fig. 3, Supplementary Fig. 8, Supplementary data file 6). We identified an *NK* cluster on chromosome 5 containing *NK1*, *NK5*, *Msx*, *NK4*, *NK3*, *NK7*, *NK6*, a more distant *Lbx*, a possible highly derived *Tlx-like* gene and, intriguingly, *Hex*, which is also linked to the NK cluster in the hemichordate *Saccoglossus kowalevskii*[53] and in the cephalochordate *Branchiostoma*

*floridae*. Similar to Bilateria, the *NK2* genes were clustered separately and found on the chromosome 2 (Fig. 3, Supplementary Fig. 8-9, Supplementary data file 6). In contrast, in the earlier-branching sponges, neither ParaHox nor extended Hox cluster genes exist, and only the NK cluster is present with a single *NK2/3/4* gene, two *NK5/6/7* genes, an *Msx* ortholog, as well as possible *Hex* and *Tlx* orthologs[54], (Fig. 3, Supplementary Fig. 10). Taken together, this allows us to propose that the bilaterian-cnidarian MRCA possessed an NK-cluster on a chromosome different from the one carrying the SuperHox cluster, and a separate NK2 cluster, possibly on the same chromosome as the SuperHox cluster (Fig. 3). The hypothesized SuperHox-NK Megacluster[43], if it ever existed, must have both formed and broken apart during the time after the separation of the sponge lineage, but before the origin of the bilaterian-cnidarian ancestor (Fig. 3a, Supplementary notes). The lack of selection pressure in favor of microsynteny conservation is clearly illustrated by the comparison of the divergent Hox clusters of *Scolanthus* and *Nematostella*. Although located on homologous chromosomes, the gene order, orientation, and the number of intervening genes differs substantially between these two species (Fig. 3b). In contrast to the atomized Hox clusters of *Nematostella* and *Scolanthus*, the Hox cluster of their very distant anthozoan relative *Xenia*, is compact and contains "anterior" Hox genes *HoxA* and *HoxB* immediately next to the non-anterior Hox gene *HoxE* (Fig. 3b) Aside from the tandem duplications, this compact state likely represents the ancestral organization. Similarly, *HoxE* and *HoxB* are immediate neighbors in the genome of the jellyfish *Rhopilema*, although the *Rhopilema* Hox cluster shows some evidence of disintegration (Fig. 3b).

## Topologically associating domains are not detected in either sea anemone genome

In the past decade, high-resolution chromosome conformation capture has increased interest in topologically associating domains (TADs), recurring chromosomal-folding motifs evidenced by signals in Hi-C contact maps[55]. Flanking regions of TADs are positively correlated with CCCTC-binding factor (CTCF) binding sites. Interestingly, no CTCF ortholog has been detected in non-bilaterian animals[31,55], but previous studies of non-bilaterian animals have proposed the existence of TAD-like structures[47].

Similar to previous studies, we sought to identify putative TADs within the edwardsiid genomes. Initial inspections of the Hi-C maps of both *Scolanthus* and *Nematostella*, however, revealed that both species appeared to lack TAD-like structures (Fig. 4a; Supplementary Fig. 11) and instead the Hi-C contact frequency decayed smoothly as a function of read pair distance along the chromosomes (Supplementary Fig. 11). We quantified these findings by measuring the strength of topological boundaries, measured by insulation score, and found the topological boundaries to be weaker in the *Nematostella* genome compared to the fly (Supplementary Fig. 12, $p = 8.38 \times 10^{-18}$, Mann-Whitney *U* test) and mouse genomes ($p = 2.93 \times 10^{-55}$, Mann-Whitney *U* test). We also verified that there was a lack of bilaterian-like TADs and weak topological domain boundaries in an independent *Nematostella* chromosome-scale genome assembly and Hi-C dataset produced in parallel to ours (Supplementary Fig. 11[56]).

TAD structures are often well-conserved around gene clusters that control spatio-temporal gene expression, especially of developmental regulators, such as the TADs that separate the anterior and posterior hox genes in the mouse HoxD cluster[57]. We therefore explored whether TAD-like structures existed around Hox clusters in *Nematostella*. *Nematostella* has six Hox genes, one is on chromosome 5, while the other five Hox genes are located on chromosome 2, but in two clusters, separated by hundreds of genes between (Fig. 4). We analyzed the genomic regions surrounding both partial hox clusters but again we could not detect any significant TAD-like structure.

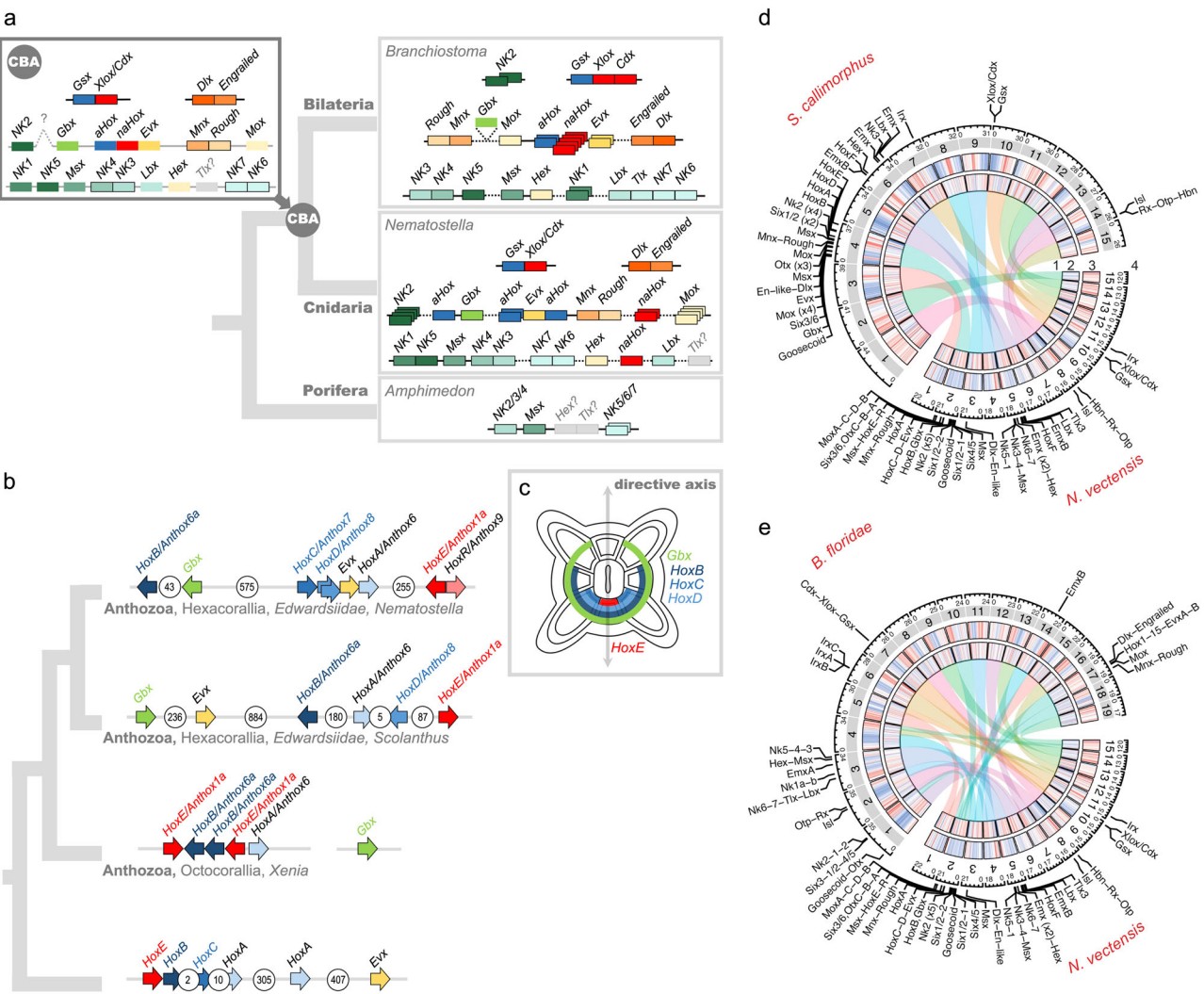

**Fig. 3 | Evolution of a selection of Antennapedia class homeobox gene clusters.**
**a** Composition of the extended Hox, ParaHox, and NK clusters in the sponge *Amphimedon*, the sea anemone *Nematostella*, in a chordate *Branchiostoma*, and the deduced cluster composition of the cnidarian-bilaterian ancestor (CBA). Grayed-out genes with question marks have uncertain orthology. Genes shown as bor-derless boxes have an uncertain position relative to neighboring cluster members. Stacked boxes represent clusters of paralogs of the indicated ancestral gene. Genes in immediate proximity are indicated by abutting boxes. Linked genes of the same class separated by 1 to 50 intervening genes are connected with solid lines, over 50, with dashed lines. Gray intergenic connectors in the CBA indicate that the distances and the number of the intervening genes between the cluster members cannot be estimated. *NK2* of the CBA may be linked to the extended Hox cluster. Since *Branchiostoma Gbx* remained unplaced in the chromosome-level assembly, its position was taken from the scaffold-level assembly of *Branchiostoma lanceolatum*[159]. A two-gene ParaHox CBA scenario is shown although a three-gene

ParaHox CBA scenario is possible based on evidence from *Scyphozoa*[21].
**b** Organization of the *Nematostella* Hox cluster in comparison to the Hox clusters of *Scolanthus*, the octocoral *Xenia* and a scyphozoan jellyfish *Rhopilema* indicates loss of microsynteny. **c** Staggered expression of *Gbx* and Hox genes along the directive axis of *Nematostella* (oral view) partially reflects the position of the genes on the chromosome. Arrows show the direction of transcription for each of the genes. The number of intervening genes is indicated in white circles. **d**, **e** Chromosomal rela-tionships, genomic content and locations of NK and extended Hox cluster and other landmark homeobox genes of (**d**) *Nematostella* and *Scolanthus* and (**e**) *Nematostella* and *Branchiostoma*. 1) Chord diagram of macrosyntenic relationships of chromosomes based on the inferred ancestral linkage groups. 2) Scaled and centered gene density relative to the respective genome (red=high, blue=low). 3) Scaled and centered density of interspersed repeat elements relative to the respective genome (red=high, blue=low). 4) Locations of the landmark homeo-box genes.

## Correlation of gene regulation with three-dimensional genome organization

We note that the *Nematostella* genome is fairly gene-dense (10 ± 4 genes per 100 kb), leaving relatively little intergenic sequence. It is conceivable that most relevant cis-regulatory elements are located in close proximity to the regulated gene. This is supported by the observation that many *Nematostella* transgenic reporter constructs faithfully mimic endogenous expression patterns with only 1–3 kb of upstream promoter sequence. Regions of open chromatin are reflec-ted by ATAC-seq peaks, many of which contain cis-regulatory regions. We therefore wished to assess whether there is a correlation between

the distance of ATAC-seq peaks and the closest gene (i.e., transcrip-tional start or end site) and the genome size. We compared several species, where chromosome assemblies and ATAC-seq data are avail-able and - as expected - we found that there is indeed a correlation between distance of ATAC-seq peaks to the next gene and the genome size of the organism. However, there is also a correlation between small genomes and the absence of clear TAD structures, for instance in *C. elegans*, sponges and cnidarians. Notably, at least one cnidarian genome, *Hydra*, is relatively large (1.27 Gb), yet does not show typical TAD structures, although smaller-scale 3D structures could be observed[58]. Thus, within bilaterians, species with small genomes may

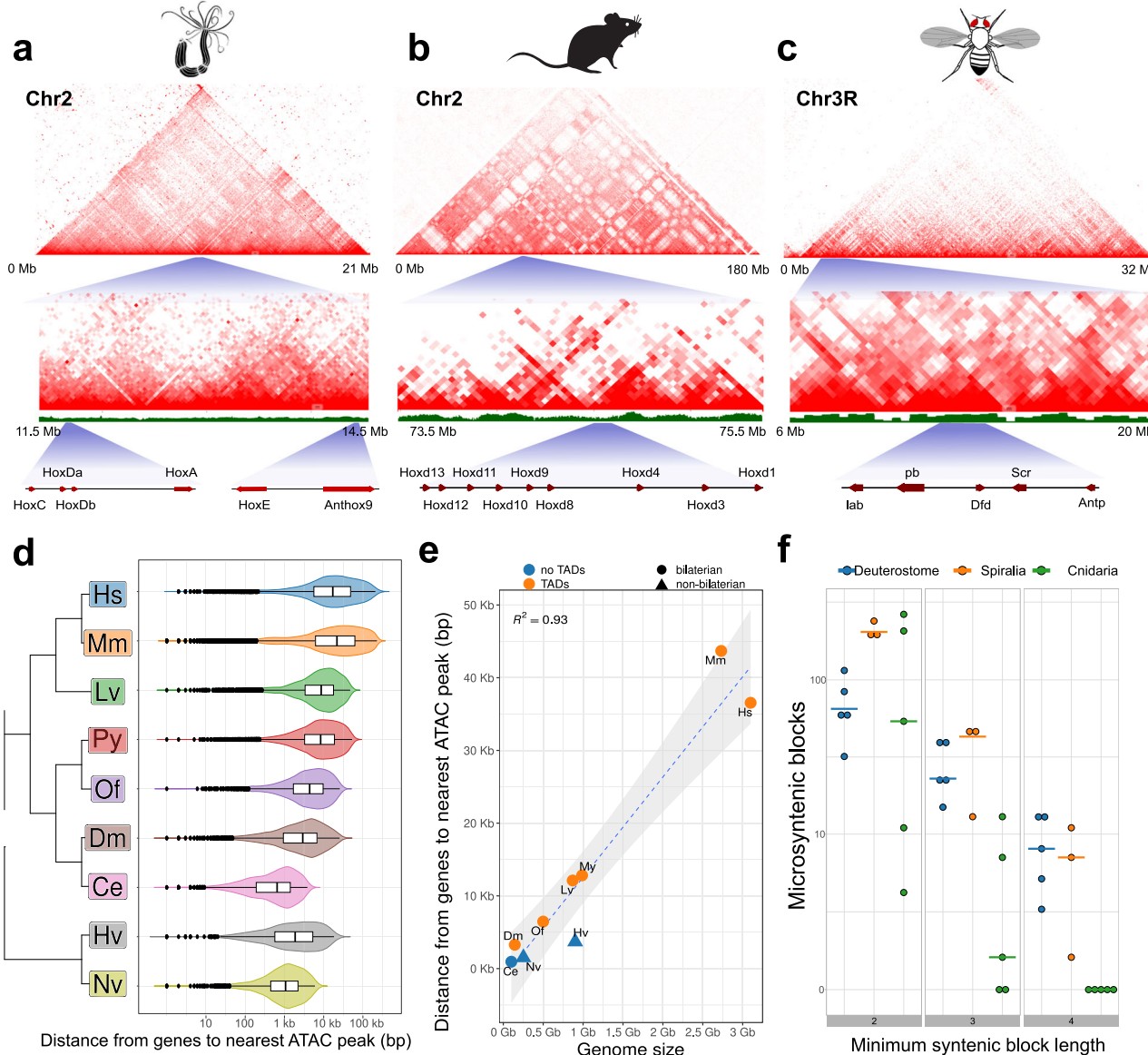

**Fig. 4 | Hi-C contact maps.** Contact maps (**a**) Nematostella Hox clusters, (**b**) Mouse HoxD cluster and (**c**) Fly Hox cluster show striking differences in the patterns of intrachromosomal interactions; **d** distribution of the distance from ATAC-seq peaks to their closest peak for human (Hs; $n = 2069$; SEM = 708.23), mouse (Mm; $n = 73,834$; SEM = 192.78), Sea urchin (*Lytechinus variegatus*, Lv; $n = 6441$; SEM = 139.49), mussel (*Patinopecten yessoensis*, Py; $n = 8831$; SEM = 132.51), polychete (*Owenia fusiformis*, Of; $n = 3245$; SEM = 107.08), fly (Dm; $n = 2811$; SEM = 77.38), *Caenorhabditis* (Ce; $n = 4043$; SEM = 14.55), *Hydra vulgaris* (Hv; $n = 2069$; SEM = 94.12) and *Nematostella vectensis* (Nv; $n = 1611$; SEM = 35.3) Vertical bars represent the median, boxes represent the interquartile range and whiskers represent 1.5× the interquartile range. Outliers are represented by dots; **e** correlation of mean gene-ATAC-seq peak distance and genome size for the same species (regression confidence interval 95%).; **f** Number of microsyntenic blocks of 2, 3, and 4 genes in cnidarians, deuterostomes and spiralians. The number of syntenic blocks falls rapidly in cnidarians compared to deuterostomes and spiralians as the synteny length increases. Horizontal bars represent the distribution median of within-clade comparisons. Mouse silhouette is from PhyloPic (https://www.phylopic.org/; Daniel Jaron, 2018).

show a tendency to lack TAD structures, while in non-bilaterians, which lack CTCF, they may not organize into stable TAD-like structures. At present, *Hydra* is the only available large non-bilaterian genome with relatively short distances between ATAC-seq peaks to the next gene.

TADs not only assure maintenance and directionality of enhancers to the regulated gene, they frequently also encompass more than one gene. In vertebrates, many TADs are conserved between species, suggesting that their maintenance is under selection pressure. This predicts that in organisms with TADs the number of microsyntenic genes is higher than in organisms lacking TADs. To test this, we compared the number of microsyntenies in different cnidarians, chordates and protostomes, roughly representing similar times of divergence. We find that, while the amount of microsyntenies consisting mostly of

two consecutive orthologous genes did not show any difference between clades, microsyntenies consisting of at least three or four orthologous genes were reduced in cnidarians. No microsyntenies of four and more genes were found in any pair of species spanning the anthozoan-medusozoan split (e.g., *Nematostella-Rhopilema*), whereas around a dozen such regions are shared among distantly branching spiralian and chordate-vertebrate ("deuterostome") genomes (e.g., 11 blocks between the scallop *Pecten maximus* bilaterian and the annelid *Streblospio benedicti*, and 14 blocks for amphioxus *Branchiostoma floridae* to chicken *Gallus gallus* split) (Fig. 4f). A similar trend was observed for microsyntenies where intervening genes were allowed to be part of the blocks. For a commonly used threshold of up to 5 intervening genes[49], *Nematostella-Rhopilema* species pair showed only

7 blocks, whereas *Pecten-Streblospio* had 82, and amphioxus to chicken had 45 microsyntenic blocks retained (Fig. 4f, Supplementary data file 12). Together, this data indicates higher retention and evolutionary exploration of longer syntenic regions in bilaterian genomes, potentially allowed for by the maintained distal topological architecture.

## Discussion

Here we report the assembly of two high quality, chromosome-level edwardsiid sea anemone genomes. In comparing them in relation to other cnidarian, bilaterian and poriferan genomes, we have illuminated several intriguing aspects about early animal chromosomal evolution, the origin of NK and extended Hox clusters, the conservation of non-coding elements and the status of topologically associated domains in the bilaterian-cnidarian MRCA. In addition, the highly improved *Nematostella* genome and manually curated gene annotations will prove to be an invaluable resource for future studies of both coding and non-coding regions, structural variants among populations and continued development of functional tools for this model organism.

Nearly all members of the extended Hox cluster were distributed among distant, isolated microsyntenic blocks on chromosome 2 of *Nematostella* (pseudo-chromosome 4 in *Scolanthus*), with the single exception of *HoxF/Anthox1*, located on chromosome 5 (Fig. 3; Supplementary notes). This indicates a lack of proximity constraint on the Hox genes in cnidarians, contrasting with the situation in Bilateria. In addition, while a staggered spatiotemporal pattern of Hox expression along the secondary, directive axis of the *Nematostella* larva and polyp can be observed[59], unlike Bilateria, there is no correlation between expression and cluster position[60]. Notably, *HoxF/Anthox1* is not only located on a different chromosome but it is also the only Hox gene expressed in the (aboral) ectoderm, while all other Hox genes are expressed in the inner endomesodermal cell layer[61,62].

The dispersed NK and extended Hox clusters may be due to the diminishment or, possible absence of higher-order chromosome organization at the level of microsynteny. In line with this, it was recently observed that the HoxD cluster boundaries in the mouse genome are marked by two TAD boundaries[57], and the cluster's intra-TAD gene order is deemed to be under selective pressure[63]. The lack of a CTCF gene in the *Nematostella* genome led us to hypothesize that the cnidarians might lack TADs, as TAD presence has been attributed to the appearance of CTCF[63]. Moreover, CTCF is absent not only in cnidarians but also in earlier branching ctenophores and sponges, which provides a possibility that the existence of TADs might represent a bilaterian-specific feature. While we were unable to detect any noticeable structure resembling the current definitions of TADs, it remains an open question as to whether larger or smaller structures, beyond the resolution of our data, could yet be detected. One study has suggested evidence for the higher-order chromosomal organization in a non-bilaterian, the sponge *Ephydatia*[47]. However, the contact maps resemble patterns we observe in our assemblies at the boundary of scaffolds or contigs, which can be the result of differential mappability from repetitive content or assembly issues. We therefore deliberately do not report any results from a TAD finder, since, after multiple rigorous rounds of manual assembly update, we can assert that the data we have generated do not qualitatively represent TAD boundaries per se, and most results would be likely false positives. While the precise definition of a TAD is still evolving[64,65], both data sets lack many characters of TADs identified in CTCF-containing genomes: hierarchical compartments, mammalian-specific "corner peaks" indicating strong interactions, and in our case, loop peaks and inter-contig compartments. This suggests that the presence of CTCF is necessary for the formation of TADs. The apparent absence of TADs in our analysis could also be explained by a higher degree of cell type variability. Therefore, we cannot exclude the possibility that performing the experiment with a more homogenous cell population, or sequencing at a higher resolution, would reveal a signal on a smaller scale.

While microsynteny analyses reveal little conservation of the local gene order in the genomes of *Nematostella* and *Scolanthus*, macrosyntenic analysis of the edwardsiid chromosomes compared to available cnidarian genomes revealed a high level of conservation. We identified a stable set of 19 ALGs across all clades of sequenced cnidarian genomes. When compared to extant genomes, we can trace a small number of recombination events from the ALGs since the common cnidarian ancestor split an estimated 580 Mya. This stands in stark contrast to the history of, for example, the 326 Mya old ancestral genome of Amniota, which is estimated to have 49 distinct units, while the karyotypes of the extant amniote taxa consist of multiple translocated segments and variable chromosomes[66]. However, far more remarkable is the macrosynteny maintained between the edwardsiids, the early branching chordate *Branchiostoma*, and the sponge *Ephydatia*. Our analyses revealed clear one-to-one, one-to-few or few-to-one conservation of the chromosome-level linkages between cnidarians, sponges and early chordates, which suggests a striking retention of macrosyntenies throughout evolution of these animal lineages. This in turn allowed us to deduce a set of 16 ALGs of the last common metazoan ancestor, which was maintained in the cnidarian-bilaterian ancestor and gave rise to the 19 ancestral cnidarian chromosomes and the 17 ancestral bilaterian chromosomes[49]. The observed conservation of macrosyntenic linkages can result from a strong selective pressure to maintain intact chromosomes during meiosis, so that only local translocations within the chromosomes, scrambling local gene order, are favored. Changes in population sizes or asexual reproduction or selfing may facilitate chromosomal unit breakages. While such global chromosomal rearrangements have been observed for some clades (most notably dipterans but also nematodes and cephalopods), it still remains unclear whether additional gene regulatory constraints may exist at the whole-chromosomal level[67]. It is tempting to speculate that the emergence of the TADs in Bilateria may have restricted local rearrangements and at the same time released the constraints on maintaining the ancestral macrosyntenies conserved all the way back to the origin of multicellular animals.

The high level of macrosynteny maintaining ancestral chromosomal blocks over hundreds of millions of years contrasts with the low level of microsynteny conservation, even between more closely related cnidarians. Despite this reshuffling of genes within chromosomes, many developmental regulators involved in axial patterning show a conserved expression pattern, suggesting that their cis-regulatory elements have been maintained. For instance, in *Nematostella* and *Hydra*, which are separated over 500 Million years, all 12 Wnt genes, *brachyury*, and *foxA* are expressed at the oral pole, while *foxQ2* and *six3* are expressed aborally[68–71]. To ensure conservation of spatio-temporal gene expression patterns cis-regulatory elements must remain located in close proximity to the respective gene even when translocated to a new genomic position. This is supported by our distance analyses of ATAC-seq peaks. By contrast, bilaterians with large genomes, such as vertebrates, enhancers can be located up to several megabases away from the gene they regulate, often with several genes in between. To maintain the gene regulation by long distant enhancers, we predicted that the microsynteny conservation should be under higher selective pressure than in non-bilaterians. Indeed, microsynteny conservation is higher in vertebrates than in cnidarians, in line with recent comparisons between skates, mouse and garfish[72]. We therefore propose a hypothetical evolutionary scenario, where close cis-regulation is ancestral to metazoans and maintained in non-bilaterians, which lack CTCF. It follows that CTCF evolved in the bilaterian common ancestor and was recruited to bind to the genome to act as an insulator and, together with cohesin, as a structural component defining recombination boundaries. This kept complex gene regulation by distant enhancers as a physical boundary for recombination events, but also acted as an insulator in TADs. Bilaterians that evolve a more compact genome would have a tendency to lose TADs and the dependence on

CTCF. This seems to be the case in the *C. elegans* autosomes[73,74] and is also predicted for other small genomes, such as in urochordates.

## Methods

### Animal care and source

*Nematostella vectensis* animals (F1 of CH2xCH6, originally collected by Cadet Hand) were cultured at 18 °C under dim light conditions and fed daily with Artemia brine shrimps. Adult male and female individuals were verified by induction of spawning in isolation[75]. Spawning was induced by a combination of white light and a temperature of 25 °C for 10 h. *Scolanthus callimorphus* animals were collected at the Île Callot, Carantec, France in the frame of the Assemble grant 227799 to U.T. After transport, they were kept in seawater at 20 °C and fed freshly hatched *Artemia salina* weekly or biweekly. Spawning could not be induced in the laboratory and the sex of the polyps was unknown. *Nematostella vectensis* is a laboratory strain since several decades. Neither *Nematostella vectensis* nor *Scolanthus callimorphus* are endangered species and they are simple invertebrates. Therefore, no ethical approval was necessary.

### Sequencing

**Short read DNA-Seq**. Genomic DNA samples were extracted from both adult male and female individual *Nematostella* adults using the DNeasy Blood and Tissue Kit (Qiagen). After purification, ~5 µg of genomic DNA was recovered from each sample. Following DNA extraction, samples were sheared and size selected for ~500 bp using a Blue Pippin Prep machine (Sage Science). Following size selection, sequencing libraries were created using a KAPA HTP Library Prep kit (Roche) and subjected to paired-end sequencing on an Illumina NextSeq 500. *Scolanthus* DNA samples for library preparation were aliquoted from high molecular weight extractions, described below.

**High molecular weight DNA extraction and library prep**. *Nematostella* high molecular weight DNA was extracted at Dovetail Genomics. Samples were quantified using Qubit 2.0 Fluorometer (Life Technologies, Carlsbad, CA, USA). The PacBio SMRTbell library (~20 kb) for PacBio Sequel was constructed using SMRTbell Template Prep Kit 1.0 (PacBio, Menlo Park, CA, USA) using the manufacturer recommended protocol. The pooled library was bound to polymerase using the Sequel Binding Kit 2.0 (PacBio) and loaded onto PacBio Sequel using the MagBead Kit V2 (PacBio). Sequencing was performed on the PacBio Sequel SMRT cell, using Instrument Control Software Version 5.0.0.6235, Primary analysis software Version 5.0.0.6236 and SMRT Link Version 5.0.0.6792, yielding 24.67 Gb over 3,050,403 subreads.

High molecular weight DNA from a single *Scolanthus callimorphus* adult animal was extracted using a modified Urea-based DNA extraction protocol[76,77]. A whole animal was flash frozen and ground with mortar and pestle. While frozen, drops of buffer UEB1 (7 M Urea, 312.5 mM NaCl, 50 mM Tris-HCl pH 8, 20 mM EDTA pH 8.1% w:v N-Lauroylsarcosine sodium salt) were added and crushed with the tissue. Tissue was incubated in a final volume of 10 mL UEB1 at RT for 10 min. Three rounds of phenol-chloroform extraction were performed, followed by DNA precipitation by addition of 0.7 volume isopropanol. The pellet was transferred to a fresh tube and washed twice in 70% EtOH and twice more in 100% EtOH, dried, and resuspended in TE buffer.

A library for PacBio sequencing was then prepared from the high molecular weight sample using the SMRTbell® Express Template Prep Kit v1. The libraries were then sequenced on a PacBio Sequel machine over 3 SMRT Cells, yielding a total of 22.85 Gb over 1,474,285 subreads. An aliquot of the same sample was used to prepare a library using the NEBNext® Ultra™ II DNA Library Prep Kit for Illumina. This was then subjected to 50 cycles of single-end sequencing in one flow cell lane using an Illumina HiSeq 2500 system.

**Chicago libraries**. Two Chicago libraries were prepared as described previously[78]. For each library, ~500 ng of HMW gDNA (mean fragment length = 100 kbp) was reconstituted into chromatin in vitro and fixed with formaldehyde. Fixed chromatin was digested with DpnII, the 5′ overhangs filled in with biotinylated nucleotides, and then free blunt ends were ligated. After ligation, crosslinks were reversed and the DNA purified from protein. Purified DNA was treated to remove biotin that was not internal to ligated fragments. The DNA was then sheared to ~350 bp mean fragment size and sequencing libraries were generated using NEBNext Ultra enzymes and Illumina-compatible adapters. Biotin-containing fragments were isolated using streptavidin beads before PCR enrichment of each library. The libraries were sequenced on an Illumina HiSeq 2500 (rapid run mode). The number and length of read pairs produced for each library was: 116 million, 2 × 101 bp for library 1; 35 million, 2 × 101 bp for library 2. Together, these Chicago library reads provided 125 × sequence coverage of the genome (1–100 kb pairs).

Chromatin was extracted from a single *Nematostella vectensis* adult male and *Scolanthus callimorphus* adult (unknown sex) nuclei using the Phase Genomics Proximo Hi-C animal protocol. After proximity ligation and purification, 16 ng and 9 ng of DNA was recovered, respectively. For library preparation 1 µl of Library Reagent 1 was added 12 PCR cycles were performed. The final library was subjected to 150 total cycles of paired-end sequencing using an Illumina NextSeq 550 machine yielding a total of 13.5 gigabases.

Hi-C sequencing, *Scolanthus callimorphus* PacBio library preparation and sequencing, Scolanthus Illumina DNA library preparation and sequencing and adult *Nematostella vectensis* RNA library preparation and sequencing was performed at the Vienna Biocenter Core Facility (VBCF) NGS Unit (https://www.viennabiocenter.org/facilities). *Nematostella vectensis* DNA size selection, library preparation, and sequencing were performed by the Molecular Biology Core at the Stowers Institute for Medical Research.

Developmental and adult *Nematostella* RNA sequencing was performed as follows. *Nematostella* were spawned and eggs were dejellied and fertilized as previously described[75]. Spawning and embryo development took place at 18 °C. Eggs and embryos from different stages were collected (300 per sample) in duplicate as indicated: eggs (within 30 min of spawn), blastula (7.5 hpf), gastrula (23.5 hpf) and planula (72 hpf). Eggs and embryos were collected in eppendorf tubes and centrifuged to a pellet at 21,000 × g for 1 min. All seawater was quickly removed and pellets were resuspended in 150 ml lysis buffer (RLT buffer supplied by the Qiagen RNeasy kit (#74104), supplemented with β-mercaptoethanol). The samples were homogenized with an electric pestle (1 min continuous drilling) and further supplemented with 200 ml of the above lysis buffer. Homogenized samples were then transferred into QIAshredder columns (Qiagen #79654) and centrifuged at 21,000 × g for 2 min. The flow throughs were supplemented with 1 ml 70% ethanol and transferred to RNeasy columns and were processed according to the Qiagen RNeasy protocol. Quality and integrity of the RNA was evaluated using the Agilent RNA 600 pico kit (Agilent Technologies) and RNA samples were stored at −80 °C until further processing. cDNA libraries were then constructed for polyA stranded sequencing. The resulting libraries were sequenced on Illumina HiSeq using paired end runs (RapidSeq- 2 × 150bp).

### Genome Assembly

Size estimates for *Nematostella vectensis* and *Scolanthus callimorphus* were derived using Genomescope[79], taking the result of the highest *k* (56 and 18) which converged under the model.

Initial assemblies based on PacBio sequencing of *Nematostella* and *Scolanthus* were generated using canu version 1.8[80] with the parameters `rawErrorRate=0.3 correctedErrorRate=0.045`.

*Nematostella* haplotigs were removed using Purge Haplotigs[81]. First, the source PacBio reads were aligned onto the canu assembly

using minimap2[82] using the parameters `-ax map-pb --secondary=no`. Following this a coverage histogram was generated using the Purge Haplotigs script `readhist`. Per the documented Purge Haplotigs protocol, lower, mid, and high coverage limits were found by manual inspection of the plotted histogram to be 12, 57, and 130, respectively. All initial contigs marked as suspect or artifactual were removed from further analysis with the Purge Haplotigs script `purge`.

Due to lower sequencing coverage of *Scolanthus*, diploid per-scaffold coverage could not be deconvolved from haploid, and therefore Purge Haplotigs could not be used. Removal of redundant contigs was performed with Redundans version 0.14a[83] using the parameters `--noscaffolding --nogapclosing --overlap 0.66`. Only contigs marked in the reduced version of the genome were used in further analysis.

The input de novo assembly, shotgun reads, and Chicago library reads were used as input data for HiRise, a software pipeline designed specifically for using proximity ligation data to scaffold genome assembly[78]. Shotgun and Chicago library sequences were aligned to the draft input assembly using a modified SNAP read mapper (http://snap.cs.berkeley.edu). The separations of Chicago read pairs mapped within draft scaffolds were analyzed by HiRise to produce a likelihood model for genomic distance between read pairs, and the model was used to identify and break putative misjoins, to score prospective joins, and make joins above a threshold. After scaffolding, shotgun sequences were used to close gaps between contigs.

## Repetitive DNA and Hi-C scaffolding

Repetitive DNA was found using two strategies. First, known repeats found in repbase[84] were searched in the assemblies using RepeatMasker[85] using the parameters `-s -align -e ncbi` in addition to `-species nematostella` for *Nematostella* and `-species edwardsiidae` for *Scolanthus*. Second, novel repeat sequences were found using RepeatModeler version 2.0[86]. After generating the repeat library, genomes' repeat regions were detected with the corresponding library using the same parameters in RepeatMasker.

Hi-C sequences were aligned to the reduced and repbase masked genomes of *Nematostella* and *Scolanthus* using `bwa mem`[87,88] using the parameters `-5SP`. For *Nematostella*, an additional candidate assembly was generated by mapping Hi-C sequences to the Chicago library scaffolded sequences using repbase masking (dovetail_standardmask) in addition to the contig-based scaffolding (contig_standardmask). Duplicate reads were marked with the `samblaster` utility[89], and duplicate, secondary and supplementary mappings were removed with `samtools`. These mappings were used to generate initial chromosomal assemblies using Lachesis[90], specifying the restriction site GATC. Assemblies were manually reviewed using Juicebox Assembly Tools version 1.11.08[91]. Candidate assemblies were compared using the nucmer aligner with default parameters and visualized using mummerplot[92]. Assemblies were converted over to Juicebox format using juicebox_scripts (https://github.com/phasegenomics/juicebox_scripts). In the case of *Scolanthus*, duplicate regions were clipped, and the resulting contigs were subjected to another round of alignment, assembly and review.

*Nematostella* scaffold correctness was assessed using REAPR[93]. *Nematostella* assembly nemVec1 was downloaded from the JGI website[1]. Sequences from the adult male and adult female (see Sequencing) were aligned to nemVec1 and the *Nematostella* genome after scaffolding with Chicago libraries using SMALT as well as the REAPR tool perfectmap using an expected insert size of 400, as determined from fragment analysis. Error-free bases and contiguity after breaking the genome were extracted from the results.

Genome and gene model set assembly and completeness was assessed using BUSCO version 3.0.2[33], using the gene set `metazoa_odb9` as the standard.

## Gene models

*Nematostella, Scolanthus* and *M. senile* sequences obtained from previous studies[34,94] and publicly available data (see Data Availability for details) were used to generate de novo assembled transcripts.

Trinity version 5.0.2[95] was run on each library using the flags `--min_contig_length 200 --min_kmer_cov 2`. For those which had a strand-specific library preparation, the flag `--SS_lib_type RF` was applied. To reduce redundancy, cd-hit version 4.6.8[96,97] was applied with the flags `-M 0 -c 1`. Transdecoder version 5.0.6[98] was used to detect open reading frames in the resulting reduced set of transcripts. Transcript abundance was quantified using salmon version 1.2.1[99] using the flags `--seqBias --useVBOpt --discardOrphansQuasi --softclip`.

For PacBio Iso-seq, 12 *Nematostella* RNA samples were collected over the course of multiple developmental stages, adult tissues and regeneration time points. For developmental stages, zygotes spawned by a single batch of wildtype colony were kept at 22 °C, and collected at 0, 24, 48, 72, and 7 dpf. Adult tissues were collected from sex-sorted, sexually mature wildtype individuals kept at 22 °C. The male and female mesenteries were harvested separately by surgically opening the body column and carefully peeling off the attached body column tissues. Adult oral discs were collected by surgical removal of tentacles as well as the attached pharyngeal regions. Regeneration was induced by amputating the oral part of a sexually mature individual at the mid-pharyngeal level. Regenerating tissues close to the wound were collected at 4 hpa and 12 hpa, respectively. All the samples were deep-frozen and lysed using TRIzol™ reagent (Invitrogen). Phenol-chloroform extraction was performed to remove undissolved mesoglea from adult tissues. Directzol™ RNA Miniprep Plus Kit (Zymo) was then used to purify total RNA from the aqueous phase. For each sample, 2 µg of total RNA with RIN > 7 was submitted to UC Berkeley for Iso-seq library construction.

RNA Libraries were sequenced at UC Berkeley using PacBio Sequel-II system. Raw subreads bams were processed and demultiplexed using PacBio's isoseq v3.2 conda pipeline. The steps include consensus generation, primer demultiplexing, polyA refinement and data clustering using default parameters. This resulted in the generation of 406,317 high quality HIFI reads and used to build Nvec200 transcriptome.

HIFI reads were mapped to the *Nematostella* genome using minimap2[82] using parameters (`-ax splice -uf --secondary=no`) to obtain the primary best alignments. Reads were then grouped and collapsed down to potential transcripts using PacBio's cDNA_Cupcake toolkit and TAMA[100]. Based on PacBio's guideline, transcripts with degraded 5' reads and have less than 10 FL counts were removed. Chimeric transcripts were then analyzed to find potential fusion genes. For reads that didn't map to the genome, de novo transcriptome assembly was performed using graph-based tool Cogent with kmer size equals 30. Cupcake and TAMA results were merged into non-redundant gene models using stringtie v2[101]. Deep RNA-seq reads from 4 developmental stages: egg, gastrula, pos-gastrula and planula were aligned to the genome using STAR v 2.7.3a[102]. Read alignments outside of the isoseq gene models were extracted with bedtools v2.29.2[103] and used for reference-guided transcriptome assembly using Stringtie2. Final gene models were obtained by merging Isoseq models and RNAseq models and manually corrected using previously cloned full length CDS from *Nematostella vectensis* in NCBI (Supplementary data file 7). Finally, transdecoder v5.5.0 was used to produce CDS annotation using a minimum protein length of 50 amino acids and prioritizing ORFs with significant similarity to any family in the PFAM database[104]. Alignment of the protein candidates to the PFAM database was done using Hmmer v3.1b2[105]. RNAseq libraries from the *N. vectensis* developmental time series were downloaded and aligned to the new genome using STAR v2.7.3a[102] with standard parameters. Mapping and

assignment efficiency was measured using featureCounts from the subread package[106] with the "-p" flag for paired-end libraries.

Evidence for *Scolanthus* gene models were taken from RNA-sequencing and repeats. *Scolanthus* RNA-seq reads (see Sequencing) were mapped to the *Scolanthus* contigs using STAR version 2.7.3a[102]. These mappings were used as evidence for intron junctions to generate putative gene models and estimating hidden Markov model parameters using BRAKER2[107,108]. Gene models were then refined using Augustus version 3.3.3[109] using extrinsic evidence from STAR splice junctions and the location of repeats from RepBase (see Genome Assembly) as counter-evidence for transcription. These models were filtered with the following criteria: 1) genes completely covered by RepeatModeler repeats (see Genome Assembly) were removed 2) predicted gene models were required to be either supported by external RNA-seq evidence as reported by Augustus or have a predicted ortholog as reported by Eggnog-mapper[110]. This resulted in a set of 24,625 gene models. Transcription factor identity was inferred by aligning the predicted protein sequences to Pfam A domains version 32.0[111] using hmmer version 3.3[105]. Transcription factor families were based on domains curated in a previous work[112].

Extended Hox cluster, NK cluster and ParaHox genes were found with BLAT[113] matches of published models[52,61,114–119] to the nv1 genome, taking the best hits. If an NVE gene model[34] corresponded to the matched genomic region, its location in the nv2 genome was then determined for macrosynteny analysis. In cases where no published gene was known, reciprocal BLAST hits between the bilaterian and cnidarian counterpart were taken as evidence for orthology.

### Divergence estimates
Single copy orthologs were detected by collecting common complete and duplicated BUSCO genes present in the *Scolanthus* and *Nematostella* genomes. Where duplicated BUSCOs were present, the transcript with the highest score was taken. This resulted in a total of 541 orthologs. BUSCOs found in genomes obtained from previous studies[2,21,22,26,27,32,41,44,47,120–123] were used to generate multiple alignments. Genes were aligned with mafft version 7.427 using the E-INS-i model and a maximum 1000 refinement iterations[124]. Alignments were trimmed using trimAl version 1.4.rev15 using the "gappyout" criteria[125]. A maximum likelihood tree was inferred using iqtree version 2.0.6, using the model finder partitioned on each gene, constrained to nuclear protein models[126]. Divergence estimates were determined using r8s version 1.8.1 using the Langley-Fitch likelihood method[127]. Age ranges were estimated by fixing the split between Bilateria and Cnidaria at 595.7 and 688.3 Mya[128].

Single copy orthologs were detected by collecting common complete BUSCO genes present in the *A. millepora*, *A. digitifera*, *E. pallida*, *M. senile*, *Scolanthus* and *Nematostella* genomes. This resulted in a total of 229 orthologs.

### Ultraconserved elements
In order to determine noncoding elements conserved between *Scolanthus* and *Nematostella*, genomes repeat-masked from both de novo and repbase repeats were blasted using NCBI BLAST+ version 2.10.0[129], using the flags -evalue 1E-10 -max_hsps 100000000 -max_target_seqs 100000000 -task megablast -perc_identity 0 -template_length 16 -penalty -2 -word_size 11 -template_type coding_and_optimal. Additionally, the -dbsize parameter was set to the estimated genome size. Candidate hits were then filtered using criteria loosely based on previous work[130]: for each high-scoring pair, a sliding window method was used to determine subsections of the alignment with at least 95 % identity, and extending these windows as long as the identity remains at this level. *Nematostella* elements mapping to more than one locus in the *Scolanthus* genome were reduced to the longest locus pair in both genomes. Elements mostly mapping to coding sequence were removed, and the

remaining elements were classified as intron or non-coding, depending on location. Recurring UCE sequences that were not identified by RepeatModeler or RepeatMasker were detected with blastclust version 2.2.26 requiring the length of hit to cover at least 90 % of either sequence for linkage.

### Macrosynteny analysis
*Branchiostoma floridae* gene models and sequences were retrieved from the published study[41]. All against all comparisons were performed with OMA standalone version 2.5[131]. Genomes were downloaded from previous studies[2,3,20,22,26,32,44–47,120–123,132–136].

Ancestral genome reconstruction was carried out using a graph based approach. In brief, genes were summarized into multi-species orthologous groups, and these comprised the nodes, and orthology groups occurring on the same chromosome or scaffold of two different species were linked together. A consensus approach to community detection based on the Leiden algorithm[137] was used to determine ancestral linkage groups from this graph. For details, see the Supplementary Notes.

### TAD sliding window analysis
Hi-C maps were generated for multiple species' genomes (g:) and reads (r:), including those of the ctenophore *Hormiphora californensis* (g: GCA_020137815.1, r: SRR13784181, SRR13784182)[138], the cnidarian *Hydra vulgaris* strain 105 (g: GCA_022113875.1, r: SRR14099165)[40], the cnidarian *Haliclystus octoradiatus* (g: GCA_916610825.1, r: ERR6745733)[139], the *Nematostella* assembly and reads presented in this manuscript (also, Hi-C reads from the Darwin Tree of Life project, ERR8571699[56]), the cnidarian *Diadumene lineata* (g: GCA_918843875.1, r: ERR6688655)[140], the fire jellyfish *Rhopilema esculentum* (g: https://doi.org/10.5524/100720[39], r: SRR11649085[21], the fly *Drosophila melanogaster* (g: assembly Release 6[122], r: SRR10512944[141], the scallop *Pecten maximus* (g: https://doi.org/10.6084/m9.figshare.10311068, r: SRR10119404)[142], the cephalochordate *Branchiostoma floridae* (g: GCA_000003815.2 Bfl_VNyyK, r: SRR12007919, SRR12059951)[41], and *Mus musculus* (g: GRCm39[143], r: SRR1771322-SRR1771324[144]).

The Hi-C maps were generated by mapping the raw paired-end Hi-C reads to the genome assemblies using chromap v0.2.3[145]. The pairs files were normalized using hicExplorer v3.6[146], and balanced using Cooler v0.8.10[147]. The insulation scores were then calculated using FanC v 0.9.23b[73] using 100 kb bin sizes. The peaks and valleys in the insulation scores were also called with FanC v0.9.24[148]. We defined the delta between each peak and valley as the transition from a highly-interacting region to a poorly-interaction region as the strength of transitions of topological boundaries. We compared the distributions of these deltas with an uncorrected Mann-Whitney U two-sided test to test for significant differences in the median values.

### Phylogenetic analysis of NK-like and SuperHox genes
Sequences of the NK class proteins were MUSCLE-aligned with default settings in MEGA11[149], and trimmed using the Automated1 setting in TrimAl (v. 1.3)[125]. Trimmed sequences were used for calculating the NJ, and ML trees. The NJ tree was calculated in MEGA11[149], and the ML tree was calculated in IQ-TREE2[126] using the automatic model selection algorithm. Sequences of the SuperHox cluster genes were aligned as above but not trimmed. NJ and ML trees were calculated as above.

### Nematostella gastrula ATAC-seq
Embryos were raised at 21 °C until gastrula stage (24 hpf). Ten embryos were collected and washed with cold PBS, then cold lysis buffer (10 mM Tris-HCl, pH 7.4, 10 mM NaCl, 3 mM MgCl2, 0.1% NP-40, 0.1% Tween-20, 0.01% Digitonin) was added. Embryos were disintegrated by pipetting and incubated in lysis buffer for 6.5 minutes on ice. Nuclei were then pelleted by centrifugation (0.5 rcf, 10 min, 4 °C). The pellet was resuspended in 50 µl tagmentation mix (5× TD buffer, 0.1% Tween-

20, 0.01% Digitonin, 2.5 µl Tn5 transposase). TD buffer and Tn5 transposase were kindly provided by the David Garfield lab. The transposition reaction was incubated at 37 °C for 30 min with agitation at 500 × *g* and then cooled down on ice for 5 min. DNA purification was carried out with the QIAquick PCR kit (QIAGEN, #28104) according to the manufacturer's instructions, with an elution in 20 µl elution buffer. PCR amplification was performed for 12 cycles according to Buenrostro[150]. A final purification step was performed with the QIAquick PCR kit (QIAGEN, #28104), elution in 15 µl EB. The quality of the library was validated with an agarose gel, from which DNA was extracted with a peqlab gel extraction kit, eluted in 20 µl EB, and stored at −20 °C. Library preparation and sequencing were done at Novogene and VBCF.

### Comparative ATAC-seq analysis

ATAC-seq peak calling was performed for the following species using publicly available datasets (see Supplementary data file 14 for details).

All peak callings were done following the Encode guideline for ATAC-seq. *Owenia* and *Nematostella* peak calls were done following the unreplicated data guidelines (https://www.encodeproject.org/pipelines/ENCPL344QWT/) while *Lytechinus* and *Petinopectin* following the replicated data guidelines (https://www.encodeproject.org/pipelines/ENCPL787FUN/). Briefly, raw reads were trimmed, aligned to their reference genome and filtered for mapq >=20. Psudoreplicates were produced for each library as well as for the merged alignment of multiple libraries and peak calling was done on the initial alignment, the merged alignment and all the pseudoreplicates. For unreplicated data, peaks were filtered based on p-signal <=0.05 and similar peak calls between the original alignment and at least one pseudoreplicate. For replicated data, final peaks were selected based on IDR scores between the merged library peaks and each replicate library.

ATAC-seq peaks for Human (hg19), mouse (mm10), *Caenohabditis* (Ce11), and *Drosophila* (dm6) were downloaded from the ChIP-atlas database (https://chip-atlas.org/[151,152]). Only embryonic stage datasets were used in this study. Repetitive regions were identified using RepeatModeler and RepeatMasker for all genomes and peaks overlapping repeat regions were removed for downstream analysis. A custom R script was used to find the closest gene for each ATAC-seq peak and measure the distances. All peaks overlapping genes were not considered in downstream analysis. Data points 1.5 times the interquartile range above the third quartile or below the first quartile were removed as outliers before plotting and regression. A simple linear regression was used to correlate genome size and mean intergenic distance for all species.

### Microsynteny analysis

Orthofinder version 2.5.4 was run to obtain orthogroups for the selected species. The microsynteny pipeline from ref. 49 was run for each of the three clades (deuterostomes, spiralians, cnidarians) separately. For this a subset of orthogroups that had at least one gene per species in a given clade (e.g., every cnidarian species had an ortholog for the cnidarian clade micro-synteny analysis) was generated, to correct for any missing genes. The micro-synteny analysis was thus based on 6016, 4832, and 6539 orthogroups for cnidarians, spiralians, and deuterostomes, respectively. For the most strict profiling of gene block length, we run the micro-synteny pipeline without allowance for any intervening genes. The resulting blocks were then filtered to remove micro-syntenies composed of paralogous genes. The resulting total numbers of micro-syntenic blocks given different minimal required lengths (at least 2, 3, or 4 genes) are listed in Supplementary data file 12.

### Reporting summary

Further information on research design is available in the Nature Portfolio Reporting Summary linked to this article.

## Code availability

A description of the tools and algorithms used for this work are described in the methods section and the supplement. Custom code used to analyze the data are available at https://github.com/nijibabulu/cnidariangenomes[153,154].

## Data availability

All raw data and assembled genomes are available via the National Center for Biotechnology Information under the accessions PRJNA667495, PRJNA1036184 and PRJNA430035. The assembled genomes can be downloaded, browsed and searched on publicly available browsers at https://simrbase.stowers.org/starletseaanemone and https://simrbase.stowers.org/wormanemone. Publicly available data that were used in this manuscript include SRR1771322-SRR1771324 [https://www.ncbi.nlm.nih.gov/bioproject/PRJNA273476] (mouse Hi-C), SRR10512944 (*D. melanogaster* Hi-C), and ERR8571699 (Wellcome Sanger *N. vectensis* Hi-C). Data necessary to reproduce the analyses together with the code is available for download via figshare [https://doi.org/10.6084/m9.figshare.24258598.v2].

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

## Acknowledgements

We thank Matthew Nicotra for providing us with the HMW DNA extraction protocol used for *Scolanthus*. We thank Robert Reischl for the photo of *Scolanthus* and Patrick R.H. Steinmetz and Hanna Kraus for the photo of *Nematostella vectensis* (Fig. 1). Special thanks to Tatiana Lebedeva for the cartoon drawings of *Nematostella* (Fig. 4). We are grateful to the Stowers Institute Molecular Biology Core facility, particularly Amanda Lawlor, Michael Peterson and Anoja Perera. This work was supported by grants of the Austrian Science Fund FWF (P24858; P21108) to U.T., support from the Stowers Institute for Medical Research to M.G. and an NIH Ruth L. Kirschstein NRSA (F32 GM131522) to E.M.H. We are also grateful for the support of the CNRS Marine Station in Roscoff and the Assemble grant 227799 to U.T. for collecting *Scolanthus*.

## Author contributions

B.Z. generated the DNA libraries for PacBio sequencing and for HiC, carried out most of the bioinformatic analyses and wrote the paper. J.D.M. generated the gene models and carried out other bioinformatics analyses. SMCR and WJF set up the SIMR base browser and carried out bioinformatic analyses, L.W., D.F., D.S., O.S., J.L.-W. carried out bioinformatic analyses. S.H., S.C., E.M.H.,. C.C., K.R., D.P., Y.M., G.G. generated experimental data and contributed to the bioinformatic analyses. MCG oversaw the project, contributed Isoseq data and edited the paper, U.T. designed the study, collected Scolanthus polyps, contributed various sequencing data, and wrote the paper, further edited by all authors.

## Competing interests

The authors declare no competing interests.
