## [Peer Review File · Nature Communications]

REVIEWERS' COMMENTS

Reviewer #1 (Remarks to the Author):

I have checked the revised version of this manuscript. I think that the authors quite convincingly responded to most of my comments and that of the author reviewers. I particularly think that the controls and extra analyses (regarding ATAC) bring more weight to their claims about the absence of TADs. I am a bit confused as the authors say in their letter that they reused ATAC-seq from previous studies in the method, they say they performed an ATAC experiment at Gastrula stage, they could have sampled more stages but it seems fine as a control.

The macrosynteny part is methodologically sound (even if there is an issue with the number of ALGs, see below), but its novelty is quite limited. I would suggest that the authors propose an equivalence between their numbering and the Simakov et al. 2022 ALG, especially as Oleg Simakov is now an author ... I also find that a 'stream/alluvial' plot as that of Simakov 2022 across the different cnidarians would help understand the pattern of chromosome fusion and evolution in cnidarians. The evolution of cnidarian karyotype is one of the most interesting parts of this manuscript and it would be nice to make it stand out visually.

The observation about the limited macrosynteny conservation is interesting but in a way expected from the clear lack of collinearity in Figure 1a between *Nematostella* and *Scolanthus*. I have a major concern about the lack of availability for the code used to conduct this analysis, a general outline is provided in the methods which is not acceptable. Actually, even if more details are provided in that regard, I think it would also be important that some of the code used for the inference of macrosynteny is made available.

Overall, I feel that this is a publishable and relatively nice manuscript, even if the findings reported (at the exception of the TAD one) are somehow confirmatory.

Supp info table of content:

1.14 -> strange character there

In the abstract:

"However, there is little information about the evolution of gene clusters, genome architectures and karyotypes during animal evolution." -> I find this to be a slight overstatement, I think there is quite an abundant literature dealing with that issue. Maybe this stands true for non-bilaterians, then, maybe saying "during early animal evolution" or something like that.

- "anthozoan Cnidaria" it is a bit weird to use both the latin and anglicised form of taxonomic groups in the same name. In general, I think you should homogenise a bit the taxonomic nomenclature in the paper.

Intro:

- L77 "TADs have been shown to act as boundaries of gene" -> can you really say that, TADs are compartments, they are limited by boundaries (CTCF) but do not co

Reviewer #3 (Remarks to the Author):

I am satisfied with the revised manuscript and responses to my review comments. The authors have done a good job responding to all reviewer comments and improved the manuscript accordingly. I agree with their decision to update the ortholog analysis to use OMA instead of reciprocal best BLAST.

I found one sentence with a typo:

Line 320: "We compared several species, where chromosome assemblies and ATACseq data are available and - as expected - we found that there is indeed a correlation between distance and of ATAC-seq peaks to the next gene and the genome size of the organism."

Remove the extra "and" before "of ATAC-seq peaks".

I have no further concerns or comments at this time on the manuscript and do not request any further revisions. I recommend that the manuscript be accepted.

RESPONSE TO REVIEWERS' COMMENTS

Reviewer #1:

Remarks to the Author:

I have checked the revised version of this manuscript. I think that the authors quite convincingly responded to most of my comments and that of the author reviewers. I particularly think that the controls and extra analyses (regarding ATAC) bring more weight to their claims about the absence of TADs. I am a bit confused as the authors say in their letter that they reused ATAC-seq from previous studies in the method, they say they performed an ATAC experiment at Gastrula stage, they could have sampled more stages but it seems fine as a control.

Thank you for your comments. We appreciate the thorough review of the manuscript.

The macrosynteny part is methodologically sound (even if there is an issue with the number of ALGs, see below), but its novelty is quite limited. I would suggest that the authors propose an equivalence between their numbering and the Simakov et al. 2022 ALG, especially as Oleg Simakov is now an author ...

We have added a table including the suggested equivalence to the supplement and referenced it in the text.

I also find that a 'stream/alluvial' plot as that of Simakov 2022 across the different cnidarians would help understand the pattern of chromosome fusion and evolution in cnidarians. The evolution of cnidarian karyotype is one of the most interesting parts of this manuscript and it would be nice to make it stand out visually.

Thank you for your comment. We appreciate the visualization in that manuscript, however, the intention of that particular plot is to illustrate the separated concepts of fusion or fusion-and-mixing events, whereas our method describes a smaller number of groups of cohabitating genes across a clade. We feel this is best represented by the Oxford/dot-plots as they are shown in the current version of the manuscript. We feel that the stream/alluvial plot can also be misleading as it suggests that the lines (=orthologous genes) are the same between all organisms. This is not the case. It is always only a pairwise comparison between two extant organisms. This is similar to the pairwise Oxford plots. Our network representation, by contrast, follows the fusion and splits of ancestral ALGs to the extant chromosomes and therefore provides a very different notion.

The observation about the limited microsynteny conservation is interesting but in a way expected from the clear lack of collinearity in Figure 1a between *Nematostella* and *Scolanthus*. I have a major concern about the lack of availability for the code used to conduct this analysis, a general outline is provided in the methods which is not acceptable. Actually, even if more details are provided in that regard, I think it would also be important that some of the code used for the inference of macrosynteny is made available.

We have added a Code Availability section containing a github repository with extensive code along with the link to the source data and descriptions of the usage to the manuscript.

Overall, I feel that this is a publishable and relatively nice manuscript, even if the findings reported (at the exception of the TAD one) are somehow confirmatory.

Thank you.

Supp info table of content:

1.14 -> strange character there

We have removed this.

In the abstract:

"However, there is little information about the evolution of gene clusters, genome architectures and karyotypes during animal evolution." -> I find this to be a slight overstatement, I think there is quite an abundant literature dealing with that issue. Maybe this stands true for non-bilaterians, then, maybe saying "during early animal evolution" or something like that.

Thank you. We have added "during early branching animal evolution" to the abstract.

- "anthozoan Cnidaria" it is a bit weird to use both the latin and anglicised form of taxonomic groups in the same name. In general, I think you should homogenise a bit the taxonomic nomenclature in the paper.

Thank you. We have reduced the usage of the clade nomenclature in favor of the anglicized form, with a few exceptions, in order to make the results more uniform.

Intro:

- L77 "TADs have been shown to act as boundaries of gene" -> can we really say that, TADs are compartments, they are limited by boundaries (CTCF) but do not constitute boundaries themselves

We have reworded this to say "The boundaries of TADs have been shown to act as barriers of gene regulation"

- L199: the gar is not a teleost, it diverged in actinopterygian before teleosts. The 3rd WGD is specific to teleosts.

Thank you for spotting this mistake. We have reworded this to say "Vertebrates, such as the early branching actinopterygian fish *Lepisosteus oculatus* and humans..."

- problem with references: L234, you mention ref 14 for 17 ALGs in bilateral, to me it sounds more like Simakov 2020 than the one mentioned. Generally, I think you should cite and reference Simakov 2022. I also think your number of 16 bilateral ALGs is not accurate, and you should provide a table to reference the equivalence with the ALGs defined in Simakov 2022 which clearly states that bilaterians have 23 ALGs

Thank you. The reference was indeed incorrect. We had intended to refer to Simakov et al 2013, which we have now corrected. In this case, we were speaking of the Bilaterian ALGs which are discussed in this particular citation. As mentioned above, we have now added a table addressing the equivalence to the metazoan ALGs.

- L318: "Cis-regulatory regions are reflected by ATAC-seq peaks, which detects open chromatin" I find this sentence a bit counterintuitive, ATAC detects open chromatin regions of which some correspond to cis-regulatory elements. The sentence gives the opposite impression.

Thank you for the comment. We have reversed the wording as suggested: "Regions of open chromatin are reflected by ATAC-seq peaks, many of which contain cis-regulatory regions."

- sometimes ATAC-seq is spelt and sometimes ATACseq, please check for consistency (ATAC-seq reads better)

We have changed the text to make this more consistent.

- L360-362, maybe this sentence could be made a bit simpler

We have updated this sentence to make it more readable.

Reviewer #3 (Remarks to the Author):

I am satisfied with the revised manuscript and responses to my review comments. The authors have done a good job responding to all reviewer comments and improved the manuscript accordingly. I agree with their decision to update the ortholog analysis to use OMA instead of reciprocal best BLAST.

I found one sentence with a typo:

Line 320: "We compared several species, where chromosome assemblies and ATACseq data are available and - as expected - we found that there is indeed a correlation between distance and of ATAC-seq peaks to the next gene and the genome size of the organism."

Remove the extra "and" before "of ATAC-seq peaks".

Thank you. We have updated this accordingly.

I have no further concerns or comments at this time on the manuscript and do not request any further revisions. I recommend that the manuscript be accepted.

Thank you. We appreciate your comments and thorough review.